# Emergent Temporal Reasoning: Distilling Chain-of-Thought for Zero-Shot Combinatorial Generalization

Jérémy Pawlus [1]  Philippe Helluy [2]  Svitlana Vyetrenko [3]

## Abstract

This work establishes a **Reasoning-Rich Distillation** framework extending prior research on time series analysis. We introduce a synthetic dataset of two time series plots with annotations and textual descriptions provided by a large pretrained foundation model, specifically `Qwen3.5-35B-A3B (Vision)` (Qwen Team, 2026), under an Answer-then-Explain strategy that justifies the ground-truth elements. The distilled smaller models are `Qwen3.5-2B (Text)` and `Qwen3.5-2B (Vision)`, and we investigate whether leveraging vision (text-plus-image input) or appending a pseudo-chain-of-thought, corresponding to the learned description obtained in the synthetic dataset, improves their performance in multiple time series analysis. Our results highlight the improvement obtained by leveraging both modalities, showcasing more efficient pairwise compositional reasoning, namely small models' robustness on multi-series analysis ($N \geq 3$) while being trained only on two-time-series examples. We further push the evaluation boundary to $N = 10$, demonstrating persistent pairwise compositional reasoning capabilities even at extreme (Out-Of-Distribution, OOD) complexity. This work demonstrates, for the first time, compositional generalization in time series analysis for 2B-parameter large language models.

[1]AxesSim, France [2]University of Strasbourg, France [3]Outsampler. Correspondence to: Jérémy Pawlus <jeremy.pawlus@math.unistra.fr>.

*Proceedings of the $43^{rd}$ International Conference on Machine Learning*, Seoul, South Korea. PMLR 306, 2026. Copyright 2026 by the author(s).

## 1. Introduction

### 1.1. Problem Statement: The Compositional Gap in Small Models

Distillation effectively transfers capabilities from large teacher models to compact students (Hinton et al., 2015; Xu et al., 2024), yet standard instruction tuning tends to mimic surface-level outputs without capturing the underlying reasoning process. Small Language Models (SLMs) are particularly limited in visual domains: efficient at extracting features from single curves, they fail to model the interactions between multiple time series ($N \geq 3$) without exponential parameter scaling.

We adopt an **operational definition** of **pairwise compositional reasoning** as *Combinatorial Generalization*: the capability of a student model, trained solely on dyadic interactions ($N = 2$), to zero-shot generalize to high-complexity, OOD environments ($N \geq 3$) by disentangling dense visual information. This capability is not explicitly supervised and cannot be achieved through simple visual feature interpolation.

We propose a shift in perspective: *Reasoning acts as a regularizer for Vision*. Forcing a model to verbalize its visual logic (Answer-then-Explain) prevents the "visual confusion" typical of small models in dense environments. We test this via **Zero-Shot Combinatorial Generalization** (Kojima et al., 2022): a student trained strictly on $N=2$ successfully generalizes to $N=5$ and $N=10$. Our ablations show that explicit reasoning (Chain-Of-Thought, CoT) (Wei et al., 2022) is the *sine qua non* for this regime: description-free students regress on dense OOD benchmarks, confirming that Answer-then-Explain is the mechanism that disentangles intersecting temporal features.

### 1.2. Our contributions

1. **Framework:** We introduce a Reasoning-Rich Distillation framework that transfers not just labels, but the causal logic of visual analysis (Answer-then-Explain) from foundational models to SLMs.

2. **Efficiency of Reason:** We demonstrate that verbalized reasoning (Chain-of-Thought) acts as a parameter-

efficient visual regularizer. Instead of scaling the vision encoder to handle noise, we show that teaching the model to articulate its visual logic is parameter-efficient (albeit inference-heavy due to token generation) and functionally superior. This "Reasoning-Rich" approach achieves parity with 35B+ foundation models using only 2B parameters, strictly by enforcing a structured verification process.

3. **Combinatorial Generalization:** We provide empirical evidence of pairwise compositional reasoning, where models trained strictly on dyadic comparisons ($N = 2$) successfully zero-shot generalize to dense multi-series environments ($N = 5$, and up to $N = 10$).

## 2. Background notions and Methodology

### 2.1. About distillation

Distillation is the process by which large language models (typically based on the Transformer architecture (Vaswani et al., 2017; Devlin et al., 2019)) transfer their knowledge to smaller language models for instruction-tuned tasks (Xu et al., 2024; Hinton et al., 2015). Distillation balances performance preservation and computational cost reduction. Beyond the classical logit-matching formulation (Hinton et al., 2015), modern instruction-level pipelines increasingly rely on *response-level* (a.k.a. black-box or sequence-level) distillation, in which the student is trained to reproduce the teacher's *generated outputs* rather than its internal distribution (Xu et al., 2024). This is the regime we operate in: the student does not see the teacher's logits, but learns from text it produced—specifically the natural-language rationale generated by the 35B teacher under the Answer-then-Explain prompt. Although the classification fields are deterministic ground truth, the rationale is an *emergent, teacher-specific artefact* (its vocabulary, structure, and visual–numeric bridges are not prescribed by any external oracle) and is therefore a legitimate distillation signal in the response-level sense, even though the training loss is ordinary Supervised Fine Tuning (SFT) cross-entropy over the teacher's token sequence.

### 2.2. Synthetic dataset generation and annotation

Following prior work, we generate an annotated dataset of two time series corresponding to Ornstein-Uhlenbeck stochastic processes (Byrd, 2019):

$$r_t = r_{t-1} + \kappa(\bar{r} - r_{t-1}) + u_t, \quad r_0 = 0, \qquad (1)$$

where $\bar{r}$ is a mean value of the process, $\kappa$ is a mean-reversion parameter and $u_t \sim \mathcal{N}(0, \sigma^2)$ is random noise added to the time series at each time step $t$.

Each training instance comprises a pair of such processes, featuring: (1) a dual-curve plot, (2) numerical values, (3)

deterministic ground-truth labels, and (4) a generated textual description, according to an Answer-then-Explain strategy.

O-U processes are a standard proxy for stochastic and transient patterns in finance (Wah et al., 2017; Bamford et al., 2023) and biology (Øksendal, 2003). Since Large Language Models (LLMs) achieve superior performance on visual time-series analysis than on pure text (Zhou & Yu, 2025), we use a multimodal pipeline: the reasoning-enhanced Vision Language Model (VLM) `Qwen3.5-35B-A3B (Vision)` produces a natural-language justification that validates ground-truth labels through visual and mathematical evidence, given the deterministic labels plus the exact numerical statistics from which they were computed (Answer-then-Explain). This explicit grounding minimizes hallucination; the generated *explanation* field serves as the distillation target alongside the classification labels. See Appendix B for a typical annotation.

Following this protocol and randomly sampling the parameters $(\kappa, \bar{r}, \sigma)$, we introduce 2,200 time series-image-annotation samples. Then we allocate 2000 samples for training, and 200 samples for testing/validation on the dyadic ($N = 2$) task. For the subsequent pairwise compositional reasoning assessment ($N \geq 3$), we generate a separate evaluation set of 500 samples per configuration.

#### 2.2.1. DATASET DIVERSITY AND GENERATION PARAMETERS

To ensure robust generalization capabilities, we employ a diversified generation strategy combining stochastic Ornstein-Uhlenbeck (O-U) processes with deterministic "Challenge Generators". The dataset distribution ensures coverage of topological and visual complexities: 40% Multiple Intersections, 30% Pinch Challenges, 20% High Noise, and 10% Boundary/Clear cases. The specific generation parameters are detailed below:

**Global Parameters** All time series have a fixed length of $L = 128$ points and are normalized to the integer range $[0, 99]$. The visual representation uses a fixed color palette (e.g., Blue/Orange for $N = 2$).

**Stochastic O-U Configurations** We vary the O-U parameters to create distinct topological scenarios:

- **Standard (Random):** $\kappa \in [0.05, 0.15]$, $\sigma = 2.0$, with random means $\bar{r} \in [30, 70]$. This generates natural textures.

- **Controlled "None":** Forces separation by fixing $\sigma = 5.0$ and setting means apart ($|\bar{r}_1 - \bar{r}_2| \geq 40$).

- **Controlled "Multiple":** Forces "braided" behavior by setting identical means ($\bar{r}_1 = \bar{r}_2$) with high volatility ($\sigma = 5.0$).

- **Targeted Intersections:** We use rejection sampling to force intersections in specific zones (Beginning: $i < 42$, Middle: $43 - 85$, End: $i > 86$).

**Deterministic Challenge Generators** To test fine-grained visual reasoning, we include additive signal-plus-noise scenarios not based on the O-U formulation:

- **Pinch (The "Trap"):** Parallel signals with a narrow gap (8-15 units) and medium noise ($\sigma \approx 0.5 \times$ gap). Tests the ability to detect non-crossing proximity (e.g., detecting a gap of 1 pixel).

- **High Noise:** Signal-to-Noise ratio decreased to $< 1.0$ (Noise $\sigma \in [8.0, 12.0]$). Tests filtering capabilities where the global structure (e.g., X-cross) is hidden behind heavy fluctuations.

- **Boundary Cases:** Scenarios with dominance ratios exactly at the cut-off (e.g., 74% vs 76% dominance). Tests the precision of the 75% logical rule.

### 2.3. Evaluation metrics

Given the inherent noise in Natural Language Inference (NLI)-based metrics (Williams et al., 2017), we prioritize **Field-Specific Exact Match**: outputs are constrained to a closed vocabulary (e.g., *increasing*, *beginning*) and compared strictly against deterministic ground truth. The **Average Accuracy** $\mathcal{A} = \frac{1}{N} \sum_{i=1}^{N} \frac{1}{K} \sum_{j=1}^{K} \mathbb{I}(\hat{y}_{ij} = y_{ij})$ is the mean exact-match score over $K$ attributes and $N$ samples; its negative log $\mathcal{L}_{\text{acc}} = -\log \mathcal{A}$ is used for training-dynamics plots to align with the training/validation losses. Description quality is treated as a latent variable, measured only through downstream classification accuracy.

### 2.4. Input data and prompts

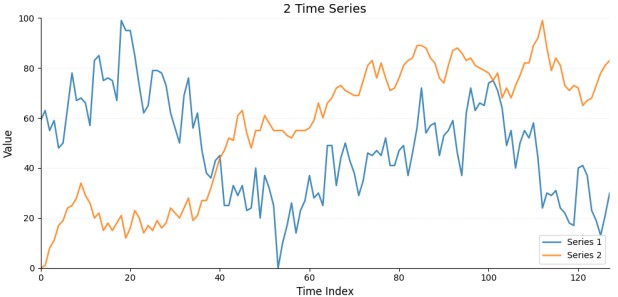

*Figure 1.* Sample two time series image input to `Qwen3.5-35B-A3B (Vision)` during synthetic data generation and `Qwen3.5-2B (Vision)` during training.

The *Teacher* prompt (Appendix F.1) applies a "Visual-First, Math-Verified" strategy: visual description of trends, extrema and interactions, verified by pre-computed statistics,

under strict vocabulary constraints. The *Student* prompt (Appendix F.2) mirrors this structure over trend, extrema, relative-position and intersection checks. The Text-Only student uses a variant relying only on numerical values; the Description-Free baseline omits the explanation field. A sample Teacher annotation is reproduced in Appendix B; the rationale is concatenated with ground-truth labels to form the SFT target.

### 2.5. Distillation Process

We distill time series reasoning into `Qwen3.5-2B (Text)` and `Qwen3.5-2B (Vision)` (Qwen Team, 2026) via supervised fine-tuning (SFT; the resulting checkpoints are tagged *Fine-Tuned (FT)* in the result tables), with a Description-Free baseline trained on classification targets only. To mitigate tokenization artifacts on floating-point inputs (Spathis & Kawsar, 2023), we scale all values to $[0, 99]$ integers following (Yuan et al., 2023).

## 3. Experimental results and evaluation

### 3.1. Training Details

All students are fine-tuned with LoRA (Low-Rank Adaptation) (Hu et al., 2021) (rank $r$=128, $\alpha$=128, dropout 0.1) using AdamW, learning rate $1\times10^{-4}$, cosine schedule, warmup ratio 0.03, weight decay 0.1, effective batch size 8, on 3×AMD MI210 in bf16. We train for up to 15 epochs with NEFTune (Noisy Embedding Fine-Tuning) noise 10 and select checkpoints before validation-loss divergence by monitoring compositional accuracy every epoch. Full hyperparameters, validation-loss epochs and training dynamics are reported in Appendix A.

### 3.2. Dyadic Analysis Results ($N = 2$)

*Table 1.* Two time series-analysis results (Detailed Breakdown)

| Configuration | Variant | S1 Trend | S2 Trend | S1 Max | S1 Min | S2 Max | S2 Min | Rel Pos | Intersect | Macro | Best Epoch |
|---|---|---|---|---|---|---|---|---|---|---|---|
| **Qwen3.5-35B-A3B (Text)** | - | 77.20 | 76.80 | 67.60 | 84.80 | 78.40 | 80.80 | 82.00 | 69.20 | 77.10 | - |
| **Qwen3.5-35B-A3B (Vision)** | - | 71.60 | 82.40 | 85.60 | 88.00 | 91.60 | 87.20 | 90.00 | 90.00 | 85.80 | - |
| **Qwen3.5-2B (Vision)** | + Description (Fine-tuned) | 91.20 | 90.00 | 97.20 | 96.00 | 96.00 | 95.20 | 88.40 | 92.80 | 93.35 | 5 |
| **Qwen3.5-2B (Vision)** | + No Description (Fine-tuned) | 92.00 | 93.20 | 98.80 | 97.20 | 98.40 | 98.40 | 94.40 | 93.60 | 95.75 | 4 |
| **Qwen3.5-2B (Text)** | + Description (Fine-tuned) | 87.60 | 88.80 | 95.60 | 94.80 | 93.60 | 96.00 | 83.20 | 87.60 | 90.90 | 7 |
| **Qwen3.5-2B (Text)** | + No Description (Fine-tuned) | 97.20 | 96.40 | 99.20 | 98.00 | 99.60 | 99.20 | 89.20 | 92.40 | 96.40 | 4 |
| **Qwen3.5-2B (Text)** | (Base) | 45.20 | 53.60 | 44.00 | 42.80 | 44.00 | 47.20 | 89.20 | 79.20 | 55.65 | - |
| **Qwen3.5-2B (Vision)** | (Base) | 70.00 | 79.60 | 68.00 | 59.20 | 60.00 | 68.00 | 49.20 | 80.80 | 66.85 | - |

Fine-tuning via distillation lifts macro scores from 35–67% (base) to $\geq 90\%$ for all variants, with the 2B students consistently matching or exceeding their 35B teachers on this in-distribution task. Interestingly, the *No Description* variants slightly edge the *+Description* ones at $N$=2: the pseudo-CoT does not help when the dyadic task is already linearly separable. Its value emerges only under out-of-distribution stress (Sec. 3.3).

### 3.3. Pairwise Compositional Reasoning Results

**Setup.** The evaluation set contains 500 multi-series plots per $N \in \{3, 4, 5, 10\}$, sharing the training visual format (e.g., Fig. 2; $N=10$ example in Appendix B). Each query targets a single pair (e.g., Series 3 vs. Series 4) via a CoT prompt (Appendix F.4) covering two high-order tasks: *Relative Positioning* and *Intersection Localisation*. Numerical values for all series are appended; outputs are scored by Field-Specific Exact Match.

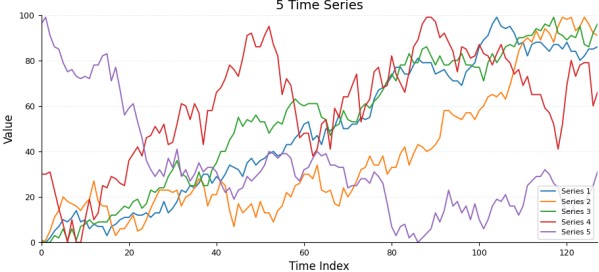

*Figure 2.* Example $N=5$ input. The prompt asks for a specific pair; the other curves act as visual distractors.

**Results.** Table 2 reports macro scores aggregated over three OOD evaluation seeds ($\{42, 43, 44\}$). Three findings stand out: (i) the *Vision+Description* student is consistently best across all $N$, including $N=10$ ($81.70 \pm 2.61\%$); (ii) it surpasses its 35B teacher by $\sim 7$–27 points on OOD $N$ (95% CIs non-overlapping at $N=5, 10$), the gap widening with scene density; (iii) removing the description degrades the vision student by $\sim 6$ points, with non-overlapping 95% CIs at $N=3, 5$, confirming the pseudo-CoT—not the visual modality alone—as the critical enabler.

*Table 2.* Macro score (%, Field-Specific Exact Match) on the pairwise compositional reasoning benchmark across $N \in \{3, 4, 5, 10\}$, aggregated over seeds $\{42, 43, 44\}$ (mean $\pm$ std). 95% CIs and per-seed breakdowns in Appendix D. Best per column in **bold**.

| Model | Variant | $N=3$ | $N=4$ | $N=5$ | $N=10$ |
|---|---|---|---|---|---|
| Qwen3.5-2B (Vision) | +Desc., FT | **82.31 $\pm$ 0.80** | **82.04 $\pm$ 2.45** | **82.44 $\pm$ 0.92** | **81.70 $\pm$ 2.61** |
| Qwen3.5-2B (Vision) | NoDesc., FT | 76.05 $\pm$ 1.14 | 75.14 $\pm$ 2.54 | 75.94 $\pm$ 1.54 | 75.87 $\pm$ 3.05 |
| Qwen3.5-2B (Text) | +Desc., FT | 59.24 $\pm$ 0.70 | 59.93 $\pm$ 0.76 | 61.68 $\pm$ 1.55 | 61.27 $\pm$ 2.98 |
| Qwen3.5-2B (Text) | NoDesc., FT | 42.34 $\pm$ 0.33 | 42.19 $\pm$ 3.19 | 46.17 $\pm$ 1.47 | 49.12 $\pm$ 2.53 |
| Qwen3.5-35B-A3B (Vision) | – | 74.75 $\pm$ 2.18 | 74.48 $\pm$ 1.10 | 71.84 $\pm$ 0.62 | 54.96 $\pm$ 2.87 |
| Qwen3.5-35B-A3B (Text) | – | 55.51 $\pm$ 3.71 | 54.59 $\pm$ 1.92 | 52.82 $\pm$ 2.45 | 52.12 $\pm$ 2.42 |
| Qwen3.5-2B (Vision) | base | 36.43 $\pm$ 7.30 | 38.80 $\pm$ 3.89 | 39.77 $\pm$ 7.51 | 41.24 $\pm$ 6.26 |
| Qwen3.5-2B (Text) | base | 35.72 $\pm$ 0.22 | 37.05 $\pm$ 2.45 | 39.37 $\pm$ 4.64 | 39.94 $\pm$ 1.03 |

Per-seed statistics and per-attribute breakdowns are in Appendices D and C.

### 3.4. Inference Cost and Efficiency

We report wall-clock latency, completion throughput, output token count and host-process RSS on the $N$-curves benchmark (500 samples per cell, mean $\pm$ std; 95% CIs and full breakdowns in Appendix E). 2B students run locally on a single AMD MI210 in bf16; the 35B-A3B teacher is served through a managed endpoint, so its RSS reflects client-side footprint only (see Appendix E).

*Table 3.* Inference cost at $N=5$ (mean $\pm$ std over 500 samples). Full results for $N \in \{3, 4, 5, 10\}$ in Appendix E.

| Model | Variant | Latency (s) | tok/s | Tokens |
|---|---|---|---|---|
| Qwen3.5-2B (Vision) | +Desc, FT | 0.47 $\pm$ 0.04 | 30.1 $\pm$ 1.7 | 2901 $\pm$ 13 |
| Qwen3.5-2B (Vision) | NoDesc, FT | 0.47 $\pm$ 0.06 | 30.2 $\pm$ 1.8 | 2901 $\pm$ 13 |
| Qwen3.5-2B (Vision) | (Base) | 0.53 $\pm$ 0.02 | 36.5 $\pm$ 1.3 | 2906 $\pm$ 13 |
| Qwen3.5-2B (Text) | +Desc, FT | 0.49 $\pm$ 0.02 | 36.2 $\pm$ 3.9 | 2925 $\pm$ 13 |
| Qwen3.5-2B (Text) | NoDesc, FT | 0.49 $\pm$ 0.02 | 36.2 $\pm$ 3.8 | 2925 $\pm$ 13 |
| Qwen3.5-2B (Text) | (Base) | 0.53 $\pm$ 0.02 | 35.4 $\pm$ 2.5 | 2926 $\pm$ 13 |
| Qwen3.5-35B-A3B (Vision) | – | 1.91 $\pm$ 0.11 | 9.9 $\pm$ 0.7 | 2906 $\pm$ 13 |
| Qwen3.5-35B-A3B (Text) | – | 1.88 $\pm$ 0.06 | 9.8 $\pm$ 0.9 | 2926 $\pm$ 13 |

Three observations follow. (i) The 2B students are $\sim 3.8$–$4.5\times$ faster than their 35B teacher across all $N$ ($4.5\times$ at $N=10$). (ii) The pseudo-CoT leaves latency and throughput essentially unchanged versus NoDesc; the $\sim 6$-point OOD gain of Vision+Desc (Table 2) is therefore obtained *without* an inference-time tax. (iii) Vision students run $\sim 20\%$ slower than text ones (30 vs. 36 tok/s at $N=5$) due to the visual-encoder overhead—the only measurable cost for the $\sim 20$-point OOD macro gain over the text variant. Vision+Desc is thus the dominant Pareto point: matching or surpassing the 35B teacher OOD at roughly a quarter of its wall-clock per query.

### 3.5. Discussion

*Compositional reasoning or robust feature extraction?* If the model relied purely on feature extraction, performance should degrade non-linearly with the visual noise (occlusion, line density) introduced at $N=5, 10$ but absent from the dyadic training. Instead, Vision+Description maintains a mean macro $\geq 81\%$ across all $N$ (Table 2), suggesting a decoupling of *Visual Perception* (local primitives) and *Logical Reasoning* (the distilled dyadic algorithm, applied to the targeted pair). The $\sim 6$-point gap between Vision+Description and Vision-NoDesc—persistent across $N$ and with non-overlapping 95% CIs at $N=3$ and $N=5$—isolates the pseudo-CoT contribution from the visual modality alone, indicating that verbalisation, not capacity or perception, is the critical enabler at scale.

## 4. Conclusion

We introduce *Reasoning-Rich Distillation*: a 35B VLM teacher, under an *Answer-then-Explain* prompt, transfers a verbalised dyadic procedure to a 2B student that retains macro accuracy $\geq 81\%$ up to $N=10$ despite training only on $N=2$, surpassing its 35B teacher OOD; removing the pseudo-CoT collapses this gain. Supervision *structure* thus offsets an $18\times$ parameter reduction on temporal reasoning.

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

# A. Training Details and Dynamics

*Table 4.* Full training hyperparameters and configuration for Qwen3.5-2B SFT using LoRA (Hu et al., 2021).

| Parameter | Value |
|---|---|
| *General Configuration* | |
| Base Model | Qwen3.5-2B (Vision) (or Qwen3.5-2B (Text)) |
| Precision | bf16 (BFloat16) |
| Hardware | $3 \times$ AMD MI210 |
| Training Data Units | 2,000 |
| Validation Split | 10% |
| Seed | 42 |
| *Optimization Hyperparameters* | |
| Learning Rate | $1 \times 10^{-4}$ |
| Optimizer | AdamW (Torch) |
| LR Scheduler | Cosine |
| Warmup Ratio | 0.03 |
| Weight Decay | 0.1 |
| Epochs | 15 |
| Per-Device Batch | 4 |
| Grad. Accum. | 2 steps |
| Effective Batch | 8 |
| *LoRA Configuration* | |
| Rank ($r$) | 128 |
| Alpha ($\alpha$) | 128 |
| Dropout | 0.1 |
| Bias | None |
| Target Modules | All Linear Layers |
| *Regularization & Efficiency* | |
| NEFTune Noise | 10 |
| Grad. Checkpoint | True |

*Table 5.* Epochs exhibiting minimum validation loss (fine-tuned models).

| Model Config | Variant | Best Epoch |
|---|---|---|
| Qwen3.5-2B (Vision) | Description | 5 |
| Qwen3.5-2B (Vision) | No Description | 4 |
| Qwen3.5-2B (Text) | Description | 7 |
| Qwen3.5-2B (Text) | No Description | 4 |

The following four plots report training/validation loss and accuracy for each configuration (text-only or vision, with or without description).

## A.1. Text-Only Models

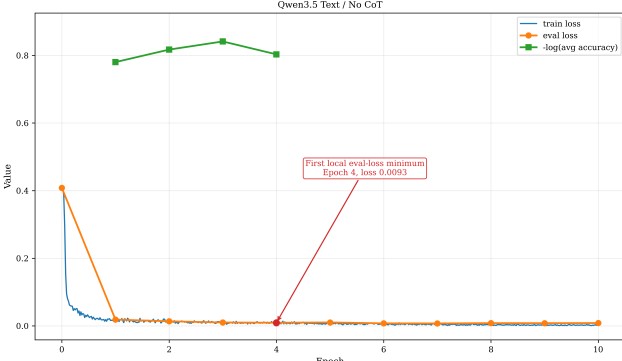

*Figure 3.* Plot containing metrics for the Text-Only Student model without acquired pseudo-CoT across all epoch.

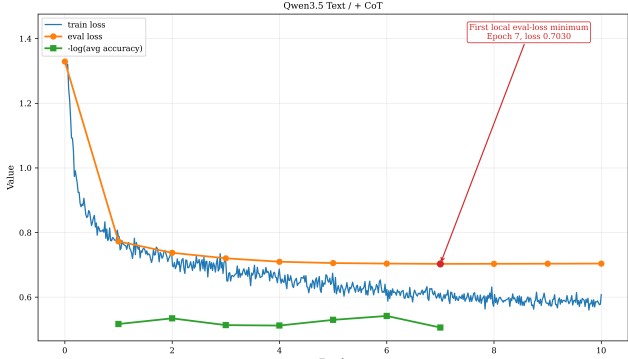

*Figure 4.* Plot containing metrics for the Text-Only Student model with acquired pseudo-CoT across all epoch.

## A.2. Vision Models

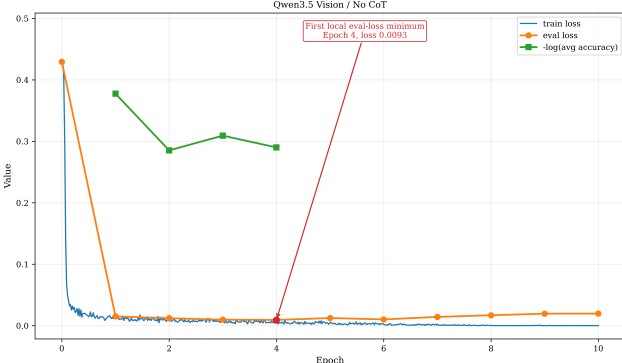

*Figure 5.* Plot containing metrics for the Vision Student model without acquired pseudo-CoT across all epoch.

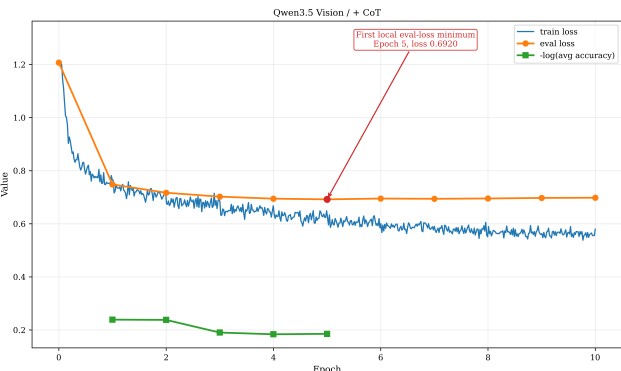

*Figure 6.* Plot containing metrics for the Vision Student model with acquired pseudo-CoT across all epoch.

## B. Sample Annotation and Additional Figures

```
ANNOTATION:
{"series 1 trend":  "decreasing", "series
2 trend":  "decreasing", "series 1 global
maximum":  "beginning", "series 1 global
minimum":  "end", "series 2 global
maximum":  "beginning", "series 2 global
minimum":  "end", "series 1 and 2 relative
position":  "1 above", "series 1 and 2
intersections locations":  "beginning",
"explanation":  "Series 1 shows a steady
downward trend with smooth fluctuations,
consistently falling from its peak at
the beginning to its lowest point at the
end.  Series 2 follows a similar pattern,
with a gradual descent marked by minor
fluctuations.  For Series 1, the start
average of 79.5 drops to 10.9 at the end,
a difference of -68.6 exceeding the 9.9
threshold, confirming a decreasing trend.
Series 2 shows a comparable decline from
78.4 to 9.4 (diff:  -68.9 > 9.9 threshold),
validating its decreasing trend as well.
Both series reach their global maximum at
index 0 (beginning), with Series 1 peaking
at 99 and Series 2 at 99, while their
global minima occur at index 125 (end),
both reaching 0.  Series 1 consistently
remains above Series 2 across 81.2% of
points, forming a clean gap with the blue
line dominating.  The only intersections
occur at indices 0 and 8, both within
the beginning section (indices 0-42),
confirming a single crossover in the early
portion of the timeline."}
```

*Figure 7.* Sample annotation generated by Qwen3.5-35B-A3B (Vision) under the Answer-then-Explain prompt. The rationale concatenated with the classification fields forms the student's SFT target.

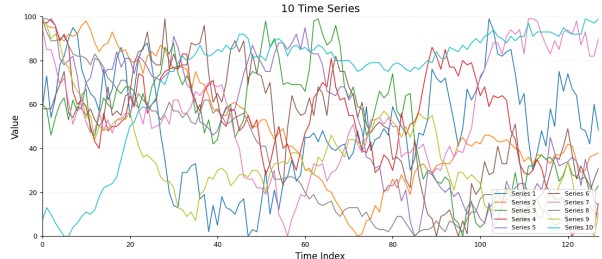

*Figure 8.* Example $N{=}10$ input used in the stress-test regime. The prompt isolates a single pair; the remaining eight curves act as visual distractors.

## C. Per-Attribute OOD Breakdown

Detailed per-attribute breakdowns and representative assessment prompts for $N \in \{3, 5, 10\}$ are available on request and will be released with the code/data artefacts.

## D. Multi-Seed Aggregated OOD Results

Tables 6–9 report the full per-$N$ multi-seed statistics summarised in Table 2. Each seed ($\{42, 43, 44\}$) controls the generation of an independent 500-sample OOD evaluation set per $N$ (fresh draws of Ornstein–Uhlenbeck parameters, noise realisations and Challenge-Generator scenarios). The 2B students are trained once with fixed training hyperparameters, and each trained checkpoint—together with the base and teacher models—is scored on the three OOD resamples. The reported variability therefore reflects *test-set resampling* (topological diversity of the generated scenes) rather than training-time stochasticity. For each cell we report mean, standard deviation and a 95% confidence interval on the mean.

*Table 6.* Pairwise compositional reasoning results aggregated over seeds 42, 43, and 44 for N=3.

| Configuration | Variant / Details | Mean Macro Score | Std | 95% CI | Best Epoch |
|---|---|---|---|---|---|
| **Qwen3.5-2B (Vision)** | **+ Description + Fine-tuned** | **82.31** | 0.80 | [80.33, 84.29] | 5 |
| Qwen3.5-2B (Vision) | + No Description + Fine-tuned | 76.05 | 1.14 | [73.22, 78.87] | 2 |
| Qwen3.5-35B-A3B (Vision) | (Base) | 74.75 | 2.18 | [69.34, 80.16] | – |
| Qwen3.5-2B (Text) | + Description + Fine-tuned | 59.24 | 0.70 | [57.50, 60.99] | 4 |
| Qwen3.5-35B-A3B (Text) | (Base) | 55.51 | 3.71 | [46.29, 64.73] | – |
| Qwen3.5-2B (Text) | + No Description + Fine-tuned | 42.34 | 0.33 | [41.53, 43.14] | 7 |
| Qwen3.5-2B (Vision) | (Base) | 36.43 | 7.30 | [18.30, 54.57] | – |
| Qwen3.5-2B (Text) | (Base) | 35.72 | 0.22 | [35.16, 36.27] | – |

*Table 7.* Pairwise compositional reasoning results aggregated over seeds 42, 43, and 44 for N=4.

| Configuration | Variant / Details | Mean Macro Score | Std | 95% CI | Best Epoch |
|---|---|---|---|---|---|
| **Qwen3.5-2B (Vision)** | **+ Description + Fine-tuned** | **82.04** | 2.45 | [75.95, 88.13] | 5 |
| Qwen3.5-2B (Vision) | + No Description + Fine-tuned | 75.14 | 2.54 | [68.83, 81.45] | 4 |
| Qwen3.5-35B-A3B (Vision) | (Base) | 74.48 | 1.10 | [71.76, 77.21] | – |
| Qwen3.5-2B (Text) | + Description + Fine-tuned | 59.93 | 0.76 | [58.04, 61.82] | 1 |
| Qwen3.5-35B-A3B (Text) | (Base) | 54.59 | 1.92 | [49.81, 59.36] | – |
| Qwen3.5-2B (Text) | + No Description + Fine-tuned | 42.19 | 3.19 | [34.26, 50.12] | 1 |
| Qwen3.5-2B (Vision) | (Base) | 38.80 | 3.89 | [29.15, 48.46] | – |
| Qwen3.5-2B (Text) | (Base) | 37.05 | 2.45 | [30.96, 43.14] | – |

*Table 8.* Pairwise compositional reasoning results aggregated over seeds 42, 43, and 44 for N=5.

| Configuration | Variant / Details | Mean Macro Score | Std | 95% CI | Best Epoch |
|---|---|---|---|---|---|
| **Qwen3.5-2B (Vision)** | **+ Description + Fine-tuned** | **82.44** | 0.92 | [80.15, 84.73] | 5 |
| Qwen3.5-2B (Vision) | + No Description + Fine-tuned | 75.94 | 1.54 | [72.10, 79.78] | 2 |
| Qwen3.5-35B-A3B (Vision) | (Base) | 71.84 | 0.62 | [70.31, 73.37] | – |
| Qwen3.5-2B (Text) | + Description + Fine-tuned | 61.68 | 1.55 | [57.84, 65.52] | 7 |
| Qwen3.5-35B-A3B (Text) | (Base) | 52.82 | 2.45 | [46.73, 58.91] | – |
| Qwen3.5-2B (Text) | + No Description + Fine-tuned | 46.17 | 1.47 | [42.52, 49.82] | 4 |
| Qwen3.5-2B (Vision) | (Base) | 39.77 | 7.51 | [21.12, 58.41] | – |
| Qwen3.5-2B (Text) | (Base) | 39.37 | 4.64 | [27.85, 50.89] | – |

*Table 9.* Pairwise compositional reasoning results aggregated over seeds 42, 43, and 44 for N=10.

| Configuration | Variant / Details | Mean Macro Score | Std | 95% CI | Best Epoch |
|---|---|---|---|---|---|
| **Qwen3.5-2B (Vision)** | **+ Description + Fine-tuned** | **81.70** | 2.61 | [75.23, 88.18] | 5 |
| Qwen3.5-2B (Vision) | + No Description + Fine-tuned | 75.87 | 3.05 | [68.30, 83.43] | 4 |
| Qwen3.5-2B (Text) | + Description + Fine-tuned | 61.27 | 2.98 | [53.86, 68.67] | 2 |
| Qwen3.5-35B-A3B (Vision) | (Base) | 54.96 | 2.87 | [47.84, 62.07] | – |
| Qwen3.5-35B-A3B (Text) | (Base) | 52.12 | 2.42 | [46.12, 58.12] | – |
| Qwen3.5-2B (Text) | + No Description + Fine-tuned | 49.12 | 2.53 | [42.83, 55.41] | 1 |
| Qwen3.5-2B (Vision) | (Base) | 41.24 | 6.26 | [25.68, 56.79] | – |
| Qwen3.5-2B (Text) | (Base) | 39.94 | 1.03 | [37.38, 42.49] | – |

# E. Detailed Performance Benchmark

Tables 10–17 report full inference statistics for every configuration across $N \in \{3, 4, 5, 10\}$. For readability, each $N$ is split into a *latency / memory* table (Latency (s) and host-process RSS after (MB)) and a *throughput / output length* table (Completion tok/s and Total tokens). For fine-tuned runs with multiple checkpoints, only the checkpoint with the best mean accuracy is retained, so each model/variant is evaluated on the same 500-sample set per $N$. Each cell shows mean ± standard deviation with the 95% confidence interval on a second line. 2B students are served locally on a single AMD MI210 in bf16; the 35B-A3B teacher is served through a managed endpoint. **The RSS column measures host-process memory only** (tokenizer, client driver, batch buffers) and therefore does not include GPU-resident weights—notably, for the 35B-A3B teacher this reflects the client-side inference driver rather than the model weights themselves, so this column is not directly comparable across teacher and student.

*Table 10.* Latency and host-process memory, N=3 (500 samples).

| Model | Variant | Latency (s) | RSS after (MB) |
|---|---|---|---|
| Qwen3.5-2B (Text) | (Base) | 0.4352 ± 0.0199 CI [0.4327, 0.4377] | 7793.90 ± 2565.22 CI [7475.92, 8111.88] |
| Qwen3.5-2B (Text) | + NoDesc, FT | 0.4082 ± 0.0226 CI [0.4054, 0.4110] | 8366.69 ± 2566.53 CI [8048.55, 8684.84] |
| Qwen3.5-2B (Text) | + Desc, FT | 0.4082 ± 0.0249 CI [0.4051, 0.4113] | 8367.18 ± 2566.52 CI [8049.03, 8685.32] |
| Qwen3.5-35B-A3B (Text) | (Base) | 1.4956 ± 0.0723 CI [1.4866, 1.5046] | 8918.17 ± 1730.65 CI [8703.64, 9132.70] |
| Qwen3.5-2B (Vision) | (Base) | 0.4303 ± 0.0209 CI [0.4277, 0.4329] | 7847.32 ± 2559.95 CI [7529.99, 8164.65] |
| Qwen3.5-2B (Vision) | + NoDesc, FT | 0.3795 ± 0.0181 CI [0.3773, 0.3817] | 8409.44 ± 2555.26 CI [8092.69, 8726.19] |
| Qwen3.5-2B (Vision) | + Desc, FT | 0.3791 ± 0.0144 CI [0.3773, 0.3809] | 8410.86 ± 2556.05 CI [8094.01, 8727.70] |
| Qwen3.5-35B-A3B (Vision) | (Base) | 1.4744 ± 0.0729 CI [1.4653, 1.4834] | 8976.17 ± 1725.49 CI [8762.28, 9190.06] |

*Table 11.* Completion throughput and total output tokens, N=3 (500 samples).

| Model | Variant | Completion tok/s | Total tokens |
|---|---|---|---|
| Qwen3.5-2B (Text) | (Base) | 43.3600 ± 3.2083 CI [42.9623, 43.7577] | 1918.98 ± 11.08 CI [1917.61, 1920.35] |
| Qwen3.5-2B (Text) | + NoDesc, FT | 43.3045 ± 4.7357 CI [42.7175, 43.8915] | 1917.84 ± 10.72 CI [1916.51, 1919.17] |
| Qwen3.5-2B (Text) | + Desc, FT | 43.4502 ± 4.5994 CI [42.8801, 44.0203] | 1917.90 ± 11.17 CI [1916.52, 1919.29] |
| Qwen3.5-35B-A3B (Text) | (Base) | 12.5121 ± 1.0901 CI [12.3770, 12.6472] | 1918.83 ± 10.90 CI [1917.48, 1920.18] |
| Qwen3.5-2B (Vision) | (Base) | 45.2479 ± 2.1321 CI [44.9837, 45.5122] | 1899.11 ± 10.77 CI [1897.78, 1900.45] |
| Qwen3.5-2B (Vision) | + NoDesc, FT | 37.1211 ± 1.6433 CI [36.9174, 37.3248] | 1893.74 ± 10.79 CI [1892.40, 1895.08] |
| Qwen3.5-2B (Vision) | + Desc, FT | 37.3728 ± 1.6555 CI [37.1676, 37.5780] | 1893.83 ± 10.81 CI [1892.49, 1895.17] |
| Qwen3.5-35B-A3B (Vision) | (Base) | 12.6013 ± 1.1236 CI [12.4620, 12.7405] | 1898.24 ± 10.85 CI [1896.89, 1899.58] |

*Table 12.* Latency and host-process memory, N=4 (500 samples).

| Model | Variant | Latency (s) | RSS after (MB) |
|---|---|---|---|
| Qwen3.5-2B (Text) | (Base) | 0.4940 ± 0.0396 CI [0.4891, 0.4989] | 7914.69 ± 2516.47 CI [7602.75, 8226.63] |
| Qwen3.5-2B (Text) | + NoDesc, FT | 0.4575 ± 0.0242 CI [0.4545, 0.4605] | 8485.69 ± 2515.58 CI [8173.86, 8797.52] |
| Qwen3.5-2B (Text) | + Desc, FT | 0.4746 ± 0.0361 CI [0.4701, 0.4791] | 8485.00 ± 2515.01 CI [8173.24, 8796.76] |
| Qwen3.5-35B-A3B (Text) | (Base) | 1.7693 ± 0.1156 CI [1.7549, 1.7836] | 8846.26 ± 1666.32 CI [8639.71, 9052.82] |
| Qwen3.5-2B (Vision) | (Base) | 0.4904 ± 0.0221 CI [0.4876, 0.4931] | 7965.44 ± 2509.03 CI [7654.42, 8276.45] |
| Qwen3.5-2B (Vision) | + NoDesc, FT | 0.4290 ± 0.0213 CI [0.4263, 0.4316] | 8530.83 ± 2506.43 CI [8220.14, 8841.53] |
| Qwen3.5-2B (Vision) | + Desc, FT | 0.4305 ± 0.0141 CI [0.4287, 0.4322] | 8530.98 ± 2506.46 CI [8220.28, 8841.68] |
| Qwen3.5-35B-A3B (Vision) | (Base) | 1.7469 ± 0.1032 CI [1.7341, 1.7597] | 8905.01 ± 1661.66 CI [8699.04, 9110.99] |

*Table 13.* Completion throughput and total output tokens, N=4 (500 samples).

| Model | Variant | Completion tok/s | Total tokens |
|---|---|---|---|
| Qwen3.5-2B (Text) | (Base) | 38.6204 ± 2.7810 CI [38.2756, 38.9651] | 2422.51 ± 12.07 CI [2421.01, 2424.00] |
| Qwen3.5-2B (Text) | + NoDesc, FT | 39.1135 ± 4.3635 CI [38.5726, 39.6544] | 2421.43 ± 12.24 CI [2419.91, 2422.95] |
| Qwen3.5-2B (Text) | + Desc, FT | 37.8889 ± 4.3765 CI [37.3464, 38.4315] | 2421.47 ± 11.99 CI [2419.98, 2422.95] |
| Qwen3.5-35B-A3B (Text) | (Base) | 10.3989 ± 1.1288 CI [10.2590, 10.5388] | 2421.86 ± 12.18 CI [2420.35, 2423.37] |
| Qwen3.5-2B (Vision) | (Base) | 39.7600 ± 1.7834 CI [39.5389, 39.9811] | 2402.52 ± 11.82 CI [2401.05, 2403.98] |
| Qwen3.5-2B (Vision) | + NoDesc, FT | 32.8882 ± 1.5549 CI [32.6955, 33.0809] | 2397.13 ± 11.80 CI [2395.67, 2398.59] |
| Qwen3.5-2B (Vision) | + Desc, FT | 33.0921 ± 1.3796 CI [32.9211, 33.2632] | 2397.28 ± 11.85 CI [2395.82, 2398.75] |
| Qwen3.5-35B-A3B (Vision) | (Base) | 10.6806 ± 0.9852 CI [10.5584, 10.8027] | 2401.67 ± 11.87 CI [2400.20, 2403.14] |

*Table 14.* Latency and host-process memory, N=5 (500 samples).

| Model | Variant | Latency (s) | RSS after (MB) |
|---|---|---|---|
| Qwen3.5-2B (Text) | (Base) | 0.5324 ± 0.0172 CI [0.5303, 0.5346] | 7962.57 ± 2434.37 CI [7660.81, 8264.33] |
| Qwen3.5-2B (Text) | + NoDesc, FT | 0.4933 ± 0.0218 CI [0.4906, 0.4960] | 8532.87 ± 2433.25 CI [8231.25, 8834.50] |
| Qwen3.5-2B (Text) | + Desc, FT | 0.4909 ± 0.0196 CI [0.4884, 0.4933] | 8533.37 ± 2433.17 CI [8231.76, 8834.98] |
| Qwen3.5-35B-A3B (Text) | (Base) | 1.8827 ± 0.0575 CI [1.8755, 1.8898] | 8947.76 ± 1664.41 CI [8741.44, 9154.08] |
| Qwen3.5-2B (Vision) | (Base) | 0.5344 ± 0.0166 CI [0.5323, 0.5364] | 8077.22 ± 2465.59 CI [7771.58, 8382.85] |
| Qwen3.5-2B (Vision) | + NoDesc, FT | 0.4675 ± 0.0647 CI [0.4595, 0.4756] | 8638.28 ± 2461.15 CI [8333.20, 8943.36] |
| Qwen3.5-2B (Vision) | + Desc, FT | 0.4723 ± 0.0449 CI [0.4668, 0.4779] | 8640.70 ± 2462.85 CI [8335.41, 8945.99] |
| Qwen3.5-35B-A3B (Vision) | (Base) | 1.9086 ± 0.1141 CI [1.8945, 1.9228] | 9007.45 ± 1659.93 CI [8801.69, 9213.21] |

*Table 15.* Completion throughput and total output tokens, $N=5$ (500 samples).

| Model | Variant | Completion tok/s | Total tokens |
|---|---|---|---|
| Qwen3.5-2B (Text) | (Base) | $35.4057 \pm 2.4707$ CI [35.0994, 35.7120] | $2926.15 \pm 13.06$ CI [2924.53, 2927.77] |
| Qwen3.5-2B (Text) | + NoDesc, FT | $36.2337 \pm 3.7508$ CI [35.7687, 36.6986] | $2925.22 \pm 13.11$ CI [2923.60, 2926.85] |
| Qwen3.5-2B (Text) | + Desc, FT | $36.2471 \pm 3.9291$ CI [35.7600, 36.7341] | $2925.14 \pm 13.13$ CI [2923.51, 2926.77] |
| Qwen3.5-35B-A3B (Text) | (Base) | $9.8194 \pm 0.9056$ CI [9.7072, 9.9317] | $2925.78 \pm 13.13$ CI [2924.16, 2927.41] |
| Qwen3.5-2B (Vision) | (Base) | $36.4688 \pm 1.3184$ CI [36.3054, 36.6323] | $2906.32 \pm 12.80$ CI [2904.73, 2907.90] |
| Qwen3.5-2B (Vision) | + NoDesc, FT | $30.2404 \pm 1.7817$ CI [30.0195, 30.4613] | $2900.88 \pm 12.79$ CI [2899.29, 2902.46] |
| Qwen3.5-2B (Vision) | + Desc, FT | $30.1169 \pm 1.7363$ CI [29.9017, 30.3322] | $2901.00 \pm 12.81$ CI [2899.41, 2902.59] |
| Qwen3.5-35B-A3B (Vision) | (Base) | $9.9023 \pm 0.6551$ CI [9.8211, 9.9835] | $2905.70 \pm 12.77$ CI [2904.12, 2907.29] |

*Table 16.* Latency and host-process memory, $N=10$ (500 samples).

| Model | Variant | Latency (s) | RSS after (MB) |
|---|---|---|---|
| Qwen3.5-2B (Text) | (Base) | $0.8680 \pm 0.0587$ CI [0.8607, 0.8752] | $8372.93 \pm 2235.55$ CI [8095.81, 8650.05] |
| Qwen3.5-2B (Text) | + NoDesc, FT | $0.8107 \pm 0.1342$ CI [0.7941, 0.8273] | $8931.21 \pm 2235.84$ CI [8654.05, 9208.36] |
| Qwen3.5-2B (Text) | + Desc, FT | $0.7937 \pm 0.0343$ CI [0.7894, 0.7979] | $8926.71 \pm 2235.67$ CI [8649.58, 9203.85] |
| Qwen3.5-35B-A3B (Text) | (Base) | $3.3199 \pm 0.0933$ CI [3.3083, 3.3315] | $8926.41 \pm 1505.16$ CI [8739.83, 9112.99] |
| Qwen3.5-2B (Vision) | (Base) | $0.8471 \pm 0.0258$ CI [0.8439, 0.8503] | $8433.11 \pm 2226.24$ CI [8157.15, 8709.08] |
| Qwen3.5-2B (Vision) | + NoDesc, FT | $0.7501 \pm 0.0240$ CI [0.7471, 0.7530] | $8983.75 \pm 2230.63$ CI [8707.24, 9260.25] |
| Qwen3.5-2B (Vision) | + Desc, FT | $0.7426 \pm 0.0263$ CI [0.7393, 0.7458] | $8983.79 \pm 2230.66$ CI [8707.27, 9260.30] |
| Qwen3.5-35B-A3B (Vision) | (Base) | $3.3208 \pm 0.1118$ CI [3.3069, 3.3346] | $8987.61 \pm 1501.43$ CI [8801.49, 9173.72] |

*Table 17.* Completion throughput and total output tokens, $N=10$ (500 samples).

| Model | Variant | Completion tok/s | Total tokens |
|---|---|---|---|
| Qwen3.5-2B (Text) | (Base) | $22.4954 \pm 1.3632$ CI [22.3264, 22.6643] | $5439.81 \pm 20.98$ CI [5437.21, 5442.41] |
| Qwen3.5-2B (Text) | + NoDesc, FT | $22.6362 \pm 2.9698$ CI [22.2681, 23.0044] | $5438.52 \pm 21.12$ CI [5435.90, 5441.13] |
| Qwen3.5-2B (Text) | + Desc, FT | $23.0386 \pm 2.6327$ CI [22.7123, 23.3650] | $5438.64 \pm 20.97$ CI [5436.04, 5441.24] |
| Qwen3.5-35B-A3B (Text) | (Base) | $5.6425 \pm 0.5233$ CI [5.5777, 5.7074] | $5439.07 \pm 20.95$ CI [5436.47, 5441.67] |
| Qwen3.5-2B (Vision) | (Base) | $23.2243 \pm 1.0037$ CI [23.0999, 23.3488] | $5419.33 \pm 20.86$ CI [5416.75, 5421.92] |
| Qwen3.5-2B (Vision) | + NoDesc, FT | $19.2869 \pm 1.3853$ CI [19.1152, 19.4586] | $5414.13 \pm 20.85$ CI [5411.55, 5416.72] |
| Qwen3.5-2B (Vision) | + Desc, FT | $19.3845 \pm 0.8785$ CI [19.2756, 19.4934] | $5414.05 \pm 20.81$ CI [5411.47, 5416.63] |
| Qwen3.5-35B-A3B (Vision) | (Base) | $5.7838 \pm 0.3162$ CI [5.7446, 5.8230] | $5418.86 \pm 20.84$ CI [5416.27, 5421.44] |

# F. Input Prompts

## F.1. Teacher LLM Prompt (Synthetic Data Generation)

**TEACHER LLM PROMPT**
**Task:** You are an expert time-series analyst. You are given two time series (Series 1: Blue, Series 2: Orange). The inputs provided below consist of the Ground Truth labels, a set of **Pre-Computed Math Evidence**, and the Raw Data Context. Your job is to generate a single JSON object with only one entry named `"explanation"`, where you justify the provided Ground Truth labels using a **"Visual-First, Math-Verified"** approach.

---

**WORKFLOW & INSTRUCTIONS**
**1. The Inputs (Provided below):**

- **GROUND TRUTH:** The target labels you must justify. Do not challenge them.

- **PRE-COMPUTED EVIDENCE:** A block of rigorous mathematical statistics (averages, exact indices, thresholds). **You must QUOTE these values** to support your arguments. Do not recalculate them manually.

- **RAW DATA:** The full list of values, used only for verifying local textures or specific shapes if needed.

**2. Explanation Strategy:** For each of the 8 required items, write a justification that weaves together: 1. **Visual Observation:** Provide a rich description of the curve's character. Mention its **texture** (smooth vs. noisy), the **sharpness** of the trend (steep vs. gradual), and any salient features like sudden spikes or plateaus observed in the image. 2. **Mathematical Proof:** Quote the specific stats from the "PRE-COMPUTED EVIDENCE" block to prove it.

---

**CRITICAL CONSISTENCY RULES**
Your explanation MUST be LOGICALLY CONSISTENT with ALL ground-truth labels. Use the vocabulary tables below as HARD CONSTRAINTS.

---

**REQUIRED VISUAL VOCABULARY (Incorporate these terms where appropriate)**
To ensure high-quality visual descriptions, try to use terms from these categories when describing the plot:

- **Texture/Noise:** "Smooth", "volatile", "jagged", "erratic", "noisy", "clean", "fluctuations".

- **Shape/Dynamics:** "Linear", "curved", "plateauing", "steep ascent/descent", "gradual drift", "sharp spike", "deep valley".

- **Interaction:** "Tightly braided", "distinct parallel layers", "diverging", "converging", "entangled".

**For TRENDS (Items 1-2):**

- **increasing:** Must include: "rises", "upward", "ascends", "climbs". Forbidden: "flat", "stable", "no clear direction".

- **decreasing:** Must include: "falls", "downward", "descends", "drops". Forbidden: "flat", "stable", "no clear direction".

- **unclear:** Must include: "insignificant", "below threshold", "volatile", "fluctuates without clear direction". Forbidden: "clearly rises", "clearly falls", "strong upward", "strong downward".

**For EXTREMA POSITIONS (Items 3-6):**

- **beginning:** Must state: "index < 43", "first third", "early", "left portion". Forbidden: "middle", "center", "end", "right".

- **middle:** Must state: "43 ≤ index < 86", "central third", "center". Forbidden: "beginning", "early", "end", "late".

- **end:** Must state: "index ≥ 86", "final third", "late", "right portion". Forbidden: "beginning", "early", "middle", "center".

**For RELATIVE POSITION (Item 7):**

- **1_above:** Must include: "blue consistently above", "series 1 dominates", "> 5%" of points. Forbidden: "intertwined", "mixed", "alternating".

- **2_above:** Must include: "orange consistently above", "series 2 dominates", "> 5%" of points. Forbidden: "intertwined", "mixed", "alternating".

- **mixed:** Must include: "intertwined", "alternating", "neither dominates", "≤ 5%". Forbidden: "consistently above", "always above".

**For INTERSECTIONS (Item 8):**

- **none:** Must use: "distinct layers", "consistent gap", "never touch", "parallel". Forbidden: "cross", "intersect", "sign change", "braided".

- **beginning:** Must use: "single crossover", "one intersection", "indices 0-42", "left third". Forbidden: "multiple", "braided", "several crossings".

- **middle:** Must use: "single crossover", "one intersection", "indices 43-85", "center". Forbidden: "multiple", "braided", "several crossings".

- **end:** Must use: "single crossover", "one intersection", "indices 86-127", "right third". Forbidden: "multiple", "braided", "several crossings".

- **multiple:** Must use: "braided", "intertwined", "weaving", "multiple crossings", "several intersections". Forbidden: "single crossing", "never cross", "one intersection".

---

**Explanation Requirements (Visual-First Strategy)**
Your explanation must address all 8 items IN ORDER. For each item:

**Trends (Items 1-2):**
1. **VISUAL:** Describe the slope and texture of the curve (smooth/volatile, ascending/descending/flat).
2. **NUMERIC:** State the average of the first 40 points vs. the last 40 points.
3. **THRESHOLD CHECK:**
- Compute: `threshold = max(10% × range, 5.0)`
- If `|difference| > threshold` → justify "increasing"

or "decreasing"
- If `|difference| ≤ threshold` → justify "unclear" by stating the drift is statistically insignificant

**Global Extrema (Items 3-6):**
1. **VISUAL:** Describe where you SEE the peak/valley (e.g., "A sharp spike visible early in the timeline").
2. **NUMERIC:** State the exact index of the first occurrence.
3. **ZONE MAPPING:**
- Index 0-42 → **beginning**
- Index 43-85 → **middle**
- Index 86-127 → **end**

**Relative Position (Item 7):**
1. **VISUAL:** Describe which curve appears dominant or if they appear entangled.
2. **NUMERIC:** State the proportion of points where series 1 > series 2.
3. **THRESHOLD CHECK:**
- If proportion > 0.75 → **1_above**
- If proportion < 0.25 → **2_above**
- Otherwise → **mixed**

**Intersections (Item 8):**
1. **VISUAL:** Describe the "white space" between curves:
- Are they clearly separated (distinct layers)?
- Do they touch once (single crossover)?
- Do they weave together (braided)?
2. **NUMERIC:** Identify where `sign(series1 - series2)` changes.
3. **ZONE MAPPING:**
- No sign changes → **none**
- All sign changes in indices 0-42 → **beginning**
- All sign changes in indices 43-85 → **middle**
- All sign changes in indices 86-127 → **end**
- Sign changes in 2+ zones → **multiple**

---

**Anti-Hallucination Rules**
1. **DO NOT invent observations.** If the visual is ambiguous, defer to the numeric calculation.
2. **DO NOT contradict the ground-truth.** Your job is to JUSTIFY it, not challenge it.
3. **DO NOT use forbidden vocabulary** from the tables above.
4. **DO NOT discuss uncertainty** about the ground-truth labels themselves.

---

**Output Format**
Return EXACTLY ONE JSON object with a single key `"explanation"`:
```
{"explanation": "<Your 8-part
justification here, addressing items
in order, using required vocabulary, and
bridging visual observations with numeric
verification.>"}
```

**Formatting rules:**

- Return ONLY the JSON object (no markdown fences, no commentary before/after).

- The explanation should be 150-300 words, flowing naturally as connected sentences.

- Address all 8 items in the specified order.

- Use vocabulary from the MUST USE columns; avoid

vocabulary from the FORBIDDEN columns.

## F.2. Vision Student LLM Prompt

**VISION STUDENT LLM PROMPT**
**Task:** Analyze the provided plot containing two time series (series 1: blue curve, series 2: orange curve) and the accompanying numerical values. Generate a single JSON object that accurately describes their properties.
**Reasoning Guidelines (Visual-First Approach):**

1. **Trend Analysis (Visual Slope & Texture)**
   **Objective:** Visually determine if the curve ascends, descends, or fluctuates without direction, then verify mathematically.
   **Visual Process:** Scan from left to right. Determine the direction (ascending/descending/flat) AND the **texture** (smooth vs. volatile/noisy).
   **Verification:** Check the numerical values. Compare the first ~40 points vs the last ~40 points.

   - Label **increasing** or **decreasing** only if the difference is significant.
   - Label **unclear** if the visual drift is negligible or contradicts the noise.

2. **Global Extrema Location (Visual Peaks & Valleys)**
   **Objective:** Spot the highest peak (Global Max) and lowest valley (Global Min) relative to the chart width.
   **Visual Process:** Spot the highest and lowest geometric points relative to the chart width.
   **Verification:** Map the index $i$ (0 to 127) to the timeline:

   - **beginning**: $i < 43$
   - **middle**: $43 \leq i < 86$
   - **end**: $i \geq 85$

3. **Global Relative Position (Visual Dominance)**
   **Objective:** Determine which color visually dominates the upper region of the plot.
   **Visual Process:** Look at the "white space" or gap between the curves. Is the Blue curve consistently above the Orange one, or vice versa?
   **Verification:** Calculate the fraction of points where $s1 > s2$ vs $s2 > s1$ (ignoring crossings).

   - If Blue visually dominates (>75% of points), choose **1_above**.
   - If Orange visually dominates (>75% of points), choose **2_above**.
   - If they appear entangled or neither dominates clearly, choose **mixed**.

4. **Intersection Locations (Gap & Contact Analysis)**
   **Objective:** Identify where the distinct colored lines cross over each other.
   **Visual Process:** Identify interaction types: **Gap** (distinct), **Pinch** (near-miss), or **Cross** (intersection).
   **Verification:** Check where $s1 - s2$ changes sign and map to zones:

   - **none**: No sign changes.
   - **beginning** (0-42), **middle** (43-85), **end** (86-127).

   - **multiple**: Crossings in >1 zone (implies "braided" behavior).

5. **Explanation (Visual Narrative)**
   **Crucial Step:** Write a concise justification for the `"explanation"` key.
   **Requirement:** Connect the **Visual Appearance** (texture, shape, pinch points) to the **Mathematical Verification**.

   - Use descriptive terms: "smooth trajectory," "high volatility," "distinct gap," "pinch point," "braided."
   - Example: "The blue curve shows a volatile downward trend, confirmed by the significant drop in averages."

**Final Output (must be exactly one JSON object):**
```
<json>
{
"series 1 trend":
"<increasing|decreasing|unclear>",
"series 2 trend":
"<increasing|decreasing|unclear>",
"series 1 global maximum":
"<beginning|middle|end>",
"series 1 global minimum":
"<beginning|middle|end>",
"series 2 global maximum":
"<beginning|middle|end>",
"series 2 global minimum":
"<beginning|middle|end>",
"series 1 and 2 relative position":
"<1_above|2_above|mixed>",
"series 1 and 2 intersections locations":
"<none|beginning|middle|end|multiple>",
"explanation":
"<Visual-First description...>"
}
</json>
```
**Formatting rules:**

- Return **only** the JSON object.

## F.3. Text-Only Student LLM Prompt

**NUMERICAL STUDENT LLM PROMPT**
**Task:** Analyze the provided numerical data for two time series (Series 1 and Series 2). The data consists of two lists of 128 numerical values each. Generate a single JSON object that accurately describes their statistical properties.
**Reasoning Guidelines (Numerical Analysis):**

1. **Trend Analysis (Mathematical Slope & Volatility)**
   **Objective:** Determine if the series values generally ascend, descend, or fluctuate without direction.
   **Process:** Compare the average of the first ~40 values against the average of the last ~40 values.

   - Label **increasing** if the end average is significantly higher than the start.
   - Label **decreasing** if the end average is significantly lower than the start.
   - Label **unclear** if the difference is negligible relative to the variance (noise) of the data.

2. **Global Extrema Location (Index Mapping)**
   **Objective:** Identify the index $i$ (0 to 127) where the maximum and minimum values occur for each series.
   **Process:** Find the index of the global max and global min, then map it to these zones:

   - **beginning**: index $0 \leq i < 43$
   - **middle**: index $43 \leq i < 86$
   - **end**: index $86 \leq i \leq 127$

3. **Global Relative Position (Value Dominance)**
   **Objective:** Determine which series is numerically greater than the other for the majority of the timeline.
   **Process:** Calculate the percentage of indices where Series $1[i] >$ Series $2[i]$.

   - If Series 1 is greater $> 75\%$ of the time, choose **1_above**.
   - If Series 2 is greater $> 75\%$ of the time, choose **2_above**.
   - Otherwise, choose **mixed**.

4. **Intersection Locations (Sign Change Detection)**
   **Objective:** Identify indices where the series cross each other.
   **Process:** Detect indices where (Series $1[i] -$ Series $2[i]$) changes sign compared to the previous index. Map these crossings to zones:

   - **none**: No sign changes detected.
   - **beginning** (0-42), **middle** (43-85), **end** (86-127).
   - **multiple**: Crossings detected in more than one zone.

5. **Explanation (Analytical Summary)**
   **Crucial Step:** Write a concise justification for the `"explanation"` key.
   **Requirement:** Summarize the properties derived from the numbers.

   - Use descriptive terms based on the data: "steady upward drift," "high volatility," "distinct separation," "frequent crossing."
   - Example: "Series 1 shows a volatile downward trend, confirmed by a lower average in the final segment compared to the start."

**Final Output (must be exactly one JSON object):**
```
<json>
{
"series 1 trend":
"<increasing|decreasing|unclear>",
"series 2 trend":
"<increasing|decreasing|unclear>",
"series 1 global maximum":
"<beginning|middle|end>",
"series 1 global minimum":
"<beginning|middle|end>",
"series 2 global maximum":
"<beginning|middle|end>",
"series 2 global minimum":
"<beginning|middle|end>",
"series 1 and 2 relative position":
"<1_above|2_above|mixed>",
"series 1 and 2 intersections locations":
"<none|beginning|middle|end|multiple>",
```

```
"explanation":
"<Data-driven description...>"
}
</json>
```
**Formatting rules:**

- Return **only** the JSON object.

## F.4. Pairwise Compositional Reasoning Assessment Vision Prompt

**PAIRWISE COMPOSITIONAL REASONING ASSESSMENT VISION LLM PROMPT**
**Task:** Analyze the provided plot to describe the intersection locations between **Series 3 (green)** and **Series 4 (red)**.

______________________________________________

**Reasoning Strategy (Visual-First):**
**1. Visual Scan**

- Look at the plot image. Locate the **green curve** (Series 3) and the **red curve** (Series 4).
- Scan from left to right for "X" patterns where these two specific lines cross over each other. Ignore crossings involving other colors.

**2. Geometric Localization**

- Visually map these crossing points to the canvas width:

  - **beginning**: Left third of the image.
  - **middle**: Center third of the image.
  - **end**: Right third of the image.

**3. Numerical Verification**

- Verify the exact crossing indices using the data (where `series3[i] - series4[i]` changes sign).
- **Definitions:**

  - $i < n/3 \rightarrow$ **beginning**
  - $n/3 \leq i < 2n/3 \rightarrow$ **middle**
  - $i \geq 2n/3 \rightarrow$ **end**

- Select **multiple** if crossings appear in more than one segment, or **none** if they never cross.

______________________________________________

**Final Output (must be exactly one JSON object, nothing else):**
```
<json>
"series 3 and 4 intersections locations":
"<none|beginning|middle|end|multiple>"
</json>
```

**Input:**
[Image of the time series plot]
**Data:** You will be given the numeric values below. Use these ONLY to verify your visual observations.
**Time series values according to the following pattern:** [[0, 54, 35, 53, 5], [7, 55, 35, 53, 10], ... [88, 46, 39, 76, 87]]

### F.5. Text-Only Input Prompt for the pairwise compositional reasoning assessment

**PAIRWISE COMPOSITIONAL REASONING ASSESS-MENT TEXT-ONLY LLM PROMPT**
**Task:** Analyze the provided data/plot to describe the intersection locations between **Series 4 (red)** and **Series 5 (purple)**.

---

**Reasoning Strategy (Numerical Verification Only):**
**1. Data Analysis**

- Ignore the image. Look strictly at the provided numerical values for **Series 4** and **Series 5**.

- Calculate the difference array: $D[i] = $ Series $4[i] - $ Series $5[i]$.

**2. Sign Change Detection**

- Identify all indices $i$ where the sign of $D[i]$ is different from $D[i+1]$ (where the product $D[i] \times D[i+1] < 0$). These are the intersection points.

**3. Segmentation**

- Divide the total length $N$ into three equal segments.

- **Rules:**

  - If crossings only occur when $i < N/3 \rightarrow$ **beginning**.
  - If crossings only occur when $N/3 \leq i < 2N/3 \rightarrow$ **middle**.
  - If crossings only occur when $i \geq 2N/3 \rightarrow$ **end**.
  - If crossings occur in two or more of these segments $\rightarrow$ **multiple**.
  - If no sign changes exist $\rightarrow$ **none**.

---

**Final Output (must be exactly one JSON object, nothing else):**
```
<json>
{
"series 4 and 5 intersections locations":
"<none|beginning|middle|end|multiple>"
}
</json>
```

**Input:**
[Image of the time series plot]
**Data:** You will be given the numeric values below. Use them according to the strategy defined above.
**Formatting rules:**

- Return **only** the JSON object (no markdown fences, no commentary).

- For the answer field, output **only** the single **lowercase** word chosen from the allowed options.

- Ensure each key appears **once** and in English.

**Time series values according to the following pattern:** [[56, 91, 99, 36, 64], ..., [96, 10, 30, 81, 81]]

