# OpenReview forum: "Emergent Temporal Reasoning: Distilling Chain-of-Thought for Zero-Shot Combinatorial Generalization"
_ICML.cc/2026/Workshop/FMSD — FMSD @ ICML 2026 Poster_

### Official Review · Reviewer_4RfB · 2026-05-18
**Reasoning-Rich Distillation for OOD Time-Series Reasoning**

**Rating:** 7
**Confidence:** 4

**Review:**

**Summary**

This paper proposes a reasoning-rich distillation framework for time-series reasoning, where a large multimodal teacher generates structured explanations that are used to fine-tune compact text-only and vision-based students. Using a synthetic benchmark with training on two-series examples and testing on denser multi-series scenes, the paper shows that description supervision is especially helpful in the OOD regime, improving robustness for pairwise reasoning under distractors.

**Strengths**

- The paper studies a clear and interesting question: whether explanation-augmented distillation improves robustness for compact models in time-series reasoning, especially under multi-series visual clutter.
- The experimental pipeline is well specified. The synthetic data generation process, deterministic label construction, teacher explanation generation, fine-tuning setup, and OOD evaluation protocol are all described explicitly.
- The +Description versus No Description comparison is a meaningful ablation and provides direct evidence that rationale-style supervision helps in the multi-series OOD setting.
- +Description is weaker than No Description at N=2, but consistently stronger for N>=3 across both the vision and text students, with meaningful gaps in the OOD regime.

**Areas for Improvement**

- While the discussion argues that the gains are unlikely to come from visual feature extraction alone, I still found the mechanism behind the improvement somewhat underexplored. In particular, it remains unclear whether the descriptions teach a reusable decision procedure, or whether they mainly help the model attend to the relevant features and map them more reliably to the target labels.

**Detailed Comments**

- The work would benefit from a small qualitative analysis of the generated reasoning or explanations. For example, it would be helpful to see how +Description and No Description differ.

**Justification of Score**

This paper presents a clear and well-executed study of description-augmented distillation for time-series reasoning. The experimental setup is explicit, the `+Description` vs. `No Description` ablation is meaningful, and the `N=2` train to `N>2` test protocol provides a meaningful way to evaluate robustness under multi-series clutter. The main empirical takeaway, that description supervision helps especially in the harder OOD setting, is convincing and relevant for the workshop. My main reservation is that the mechanism behind the gain from descriptions remains somewhat unclear.

---

### Official Review · Reviewer_CrD7 · 2026-05-20

**Rating:** 4
**Confidence:** 4

**Review:**

The authors introduce Reasoning-Rich Distillation for visual/textual time-series reasoning. A 35B vision-language teacher (Qwen3.5-35B-A3B) generates Answer-then-Explain rationales for synthetic two-series Ornstein–Uhlenbeck and challenge-generated plots, and 2B Qwen student models are fine-tuned either with or without these pseudo-CoT descriptions. The authors show that students trained only on N=2 pairwise comparisons generalize zero-shot to N=3, 4, 5, and 10 multi-series plots, with the Vision+Description student achieving around 82% macro accuracy even at N=10, outperforming the 35B teacher by 7–27 points and the no-description ablation by ~6 points in this OOD regime.

Strengths:
1. The paper targets a real limitation of small multimodal models, i.e., reasoning over dense, multi-curve time-series plots rather than isolated curves.
2. The data generation process is well specified: O-U processes, controlled intersection regimes, pinch cases, high-noise cases, boundary cases, and normalized integer-valued time series. This makes the benchmark reproducible in principle and allows targeted stress tests.
3. The main empirical result is interesting. Training only on N=2 and evaluating on N up to 10 is a meaningful compositional generalization test. The Vision+Description model consistently beats Vision-NoDescription model by about 6 points on OOD multi-series settings, suggesting that rationale supervision may help beyond standard SFT.
4. The comparison between text-only, vision-only, description/no-description, base, fine-tuned, and teacher models is appreciated. It makes the central claim more testable than a single-model result.
5. The paper reports latency, token throughput, and host RSS, and acknowledges that teacher/student memory comparisons are not directly comparable because the teacher is served through an endpoint.
6. The paper is moderately novel. CoT distillation, synthetic data generation, and small-model fine-tuning are not new. The more novel part is applying rationale distillation to zero-shot combinatorial generalization in multi-time-series visual reasoning, especially the N=2 to N=10 setup. The important contribution is the controlled OOD evaluation showing robustness under increasing visual distractors.

Areas for Improvement:
1. I believe the “emergent temporal reasoning” claim is overstated. The task is highly synthetic, the label space is closed, and prompts include numerical values for all series. This makes it unclear whether the model is doing visual reasoning, numerical comparison, prompt-pattern following, or memorizing a generated decision procedure.
2. Potential leakage through numerical inputs. The OOD prompts append numerical values for all series. Since labels such as relative position and intersections are deterministic from the numbers, the improvement may not require robust visual perception. An important missing experiment is image-only evaluation, numbers-only evaluation for the vision model, and corrupted/mismatched image-vs-number evaluation.
3. The paper reports multiple OOD test seeds, but the students are trained once. This is not sufficient to establish robustness of the fine-tuning result. Training-seed variance could be substantial for LoRA SFT on only 2,000 examples.
4. Missing baselines include direct supervised labels without teacher text but with more training epochs, explicit programmatic/statistical solvers, smaller/larger students, alternative teachers, generic CoT prompts without distillation, and modern VLMs beyond Qwen. It is difficult to tell if the 6-point gap is a property of the method or of Qwen3.5-2B specifically.
5. NoDescription variants outperform Description variants. The authors explain this as CoT helping only under OOD stress, but this needs more analysis. It could indicate that rationales add noise, and the OOD gain comes from prompt alignment rather than reasoning.

Score justification: I like the direction and the controlled benchmark, but the current evidence is not yet sufficient due to possible numerical shortcuts, synthetic-only evaluation, limited statistical validation, and potentially overstated reasoning claims.

---

### Official Review · Reviewer_7kts · 2026-05-21
**Strong Methodology, Contradictory Claims**

**Rating:** 5
**Confidence:** 4

**Review:**

### Summary

The paper introduces a "Reasoning-Rich Distillation" framework to improve zero-shot combinatorial generalization in time series analysis for a 2B-parameter Small Language Model (SLM). The authors use a 35B-parameter Vision-Language Model to generate "Answer-then-Explain" annotations on synthetic 2-curve Ornstein-Uhlenbeck stochastic processes. This rationale acts as a pseudo-Chain-of-Thought (CoT) training target for the student. The study demonstrates that verbalized reasoning allows the student model to generalize from training on 2 curves to testing on up to 10 curves, effectively acting as a visual regularizer that enables strong generalization abilities far outside the training set.

### Strengths

* **Empirical Performance:** The paper presents strong empirical results demonstrating a substantial boost over all baselines. The distilled 2B vision model successfully maintains an 81.70% macro score at 10 curves.
* **Experimental Design & Ablation:** The experimental design is excellent, particularly the ablation studies that isolate the effects of different variables (Vision vs. Text, CoT vs. No-CoT).
* **Rigorous Evaluation:** The experiments are highly trustable, evaluated across multiple seeds with reported confidence intervals.
* **Efficiency Analysis:** The inclusion of latency analysis effectively demonstrates the practical value of distilling into a smaller model.

### Areas for Improvement

* **Contradictory Claims vs. Data:** The paper's central thesis is that explicit reasoning acts as a visual regularizer, claiming that "description-free students regress on dense OOD benchmarks". However, Table 2 shows the Qwen3.5-2B (Vision) NoDesc model scores 76.05% at $N=3$ and 75.87% at $N=10$. The description-free model does not regress at all. CoT provides a static accuracy boost but is demonstrably not the mechanism preventing degradation across $N$. It could be that increasing $N$ is not that much of an OOD stress test after all (see comment below). This also raised questions about whether the model displayed true “combinatorial generalization” or merely overfits to some artifacts of the synthetic data (e.g., trivially looking for certain color channels).
* **Scope of Generalization:** The claim regarding OOD generalization is somewhat narrow. The model has essentially learned to algorithmically separate two specific time series from visual clutter. The generalization scope is rather narrow, and there seems to be limited evidence how in other ways it “improves their performance in multiple time series analysis”.
* **Synthetic Data:** The paper solely relies on synthetic datasets for both training and evaluation. Real-world time series data is likely much more noisy and structurally OOD. Testing on real-world datasets would significantly strengthen the paper's overall impact.

### Detailed Comments

* **Base Models Performance under Stress Test:** 3/4 baselines did not show any performance degradation as $N$ increases. Presumably, text only models aren’t affected by increasing $N$? As for the 2B base vision model, is this because the model already lacks baseline competence at $N=3$ so additional clutter cannot degrade it further? It would help if the author offered some intuition, since this again points to the doubt that increasing $N$ is a true OOD stress test.

### Justification of Score

While the paper features rigorous analysis and a strong experimental design, I am leaning towards rejection. A central claim of the paper—that CoT is necessary to prevent regression in dense visual environments—is directly contradicted by the authors' own data, as the NoDesc baseline shows no performance degradation up to $N=10$. Furthermore, the scope of the demonstrated generalization is quite narrow and overly reliant on synthetic data. These critical flaws undermine the core narrative of the submission.